# Rosin-enabled ultraclean and damage-free transfer of graphene for large-area flexible organic light-emitting diodes

Zhikun Zhang[1],[*], Jinhong Du[1],[*], Dingdong Zhang[1], Hengda Sun[2], Lichang Yin[1], Laipeng Ma[1], Jiangshan Chen[2], Dongge Ma[2], Hui-Ming Cheng[1] & Wencai Ren[1]

The large polymer particle residue generated during the transfer process of graphene grown by chemical vapour deposition is a critical issue that limits its use in large-area thin-film devices such as organic light-emitting diodes. The available lighting areas of the graphene-based organic light-emitting diodes reported so far are usually $<1\,cm^2$. Here we report a transfer method using rosin as a support layer, whose weak interaction with graphene, good solubility and sufficient strength enable ultraclean and damage-free transfer. The transferred graphene has a low surface roughness with an occasional maximum residue height of about 15 nm and a uniform sheet resistance of $560\,\Omega$ per square with about 1% deviation over a large area. Such clean, damage-free graphene has produced the four-inch monolithic flexible graphene-based organic light-emitting diode with a high brightness of about $10,000\,cd\,m^{-2}$ that can already satisfy the requirements for lighting sources and displays.

[1] Shenyang National Laboratory for Materials Science, Institute of Metal Research, Chinese Academy of Sciences, 72 Wenhua Road, Shenyang 110016, China. [2] State Key Laboratory of Polymers Physics and Chemistry, Changchun Institute of Applied Chemistry, Chinese Academy of Sciences, 5625 Renmin Street, Changchun 130022, China. * These authors contributed equally to this work. Correspondence and requests for materials should be addressed to W.R. (email: wcren@imr.ac.cn).

Graphene is a promising material for a wide range of applications especially in next-generation flexible thin-film electronic and optoelectronic devices[1–19], such as organic light-emitting diodes (OLEDs)[12–14], and organic photovoltaic (OPV) cells[15–19], because of its two-dimensional (2D) structure, excellent electrical conductivity, high transparency, extremely high mechanical strength, good flexibility and chemical stability[20–22]. Chemical vapour deposition (CVD) on metal substrates, such as Cu, Ni and Pt, has been extensively used to grow large-area high-quality graphene films[23–28]. However, CVD-grown graphene films must be transferred from metal substrates to other substrates, such as $SiO_2$/Si and polyethylene terephthalate (PET), for electronic and optoelectronic applications[23,24,29]. Because of its atomically thin and highly transparent characteristics, a support layer has to be used to make the graphene film visible and protect it from cracking during transfer, and this layer must be removed after the graphene films have been transferred onto target substrates.

Currently, many support materials including macromolecular polymers and small organic molecules have been developed for graphene transfer[7,23,30–40]. Although macromolecular polymers can provide sufficient support to avoid graphene cracking during transfer, their strong interaction with graphene and low solubility make them difficult to be removed after the transfer. For example, polymethylmethacrylate (PMMA), the most commonly used support material[34–36], has a large adsorption energy ($E_{ad.}$) with graphene films and low solubility in any known organic solvent, and can cause local rehybridization of carbon atoms from $sp^2$ to $sp^3$ at defective sites because of their long-chain structure[34]. Therefore, a large amount of PMMA residue up to submicrometers in height is usually left on the graphene surface along with many defects after transfer[34]. Exhaustive rinsing with organic solvents such as acetone[35] and high-temperature annealing[34,36] can help remove the residue but seriously damage the graphene. Using thermal release tape and self-adhesive film as

support layer enables roll-to-roll production of large-area flexible graphene transparent conductive films; however, the transferred graphene films also suffer from big polymer residues and damages[23,30]. Although smaller organic molecules such as pentacene ($C_{22}H_{14}$)[38] and 2-(diphenylphosphory) spirofluorene ($C_{37}H_{25}OP$, SPPO1)[39] have better solubility, they either have a strong π–π interaction with graphene to hinder their removal or are too brittle to retain the integrity of the graphene during transfer.

The damages and, in particular, polymer residues introduced during transfer not only degrade the optical and electrical properties of graphene but also generate a large surface roughness[41], which may greatly limit the application of graphene in large-area thin-film devices such as OLEDs and OPV cells. For example, if used as a transparent conductive electrode (TCE), the large surface roughness often results in a high leakage current[39,42,43] and even causes a short circuit between it and the other electrode. Furthermore, the insulating polymer residues are also inhibitors for charge extraction, which significantly affects the device uniformity and accelerates failure. This situation becomes much worse when the device size is enlarged because of the increased probability of the occurrence of residue particles and damage. To meet the electrical conductivity requirement of OLEDs and OPV cells, multi-layer graphene TCEs obtained by the layer-by-layer stacking of monolayer graphene are usually used, but unfortunately, the surface roughness is multiplied. As a result, it remains a great challenge to fabricate large-area OLEDs and OPV cells using CVD-grown graphene as TCEs. As discussed in Supplementary Note 1 and summarized in Supplementary Tables 1 and 2, the available lighting area of OLEDs and the active area of OPV cells using graphene TCEs are usually less than 1 and 0.6 cm², respectively.

Here, we have found that rosin ($C_{19}H_{29}COOH$), a small natural organic molecule, is a very good support layer for the

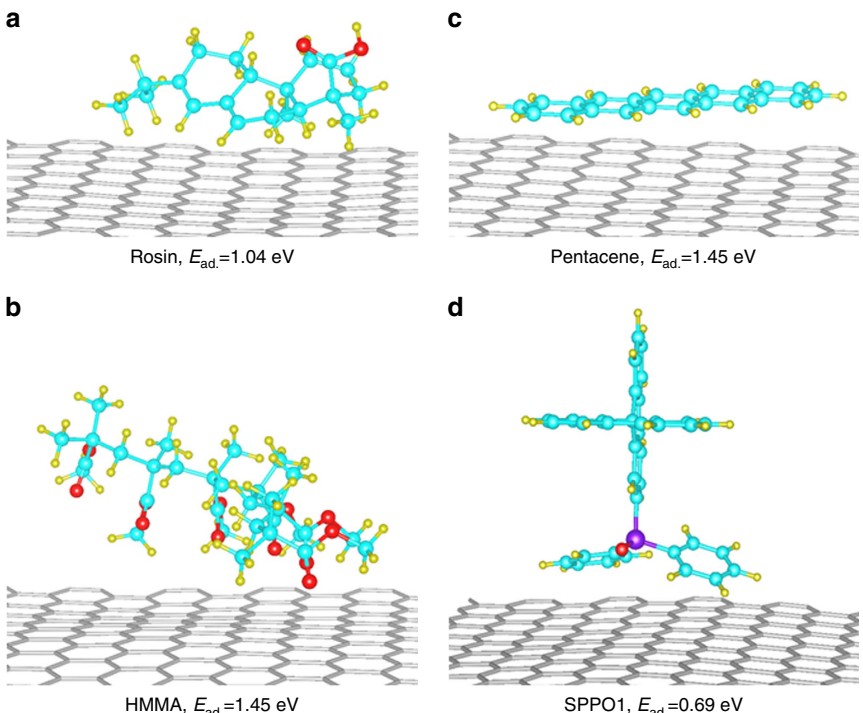

**Figure 1 | Adsorption ability of different polymeric molecules on graphene.** Schematic structures of (**a**) rosin, (**b**) HMMA, (**c**) pentacene and (**d**) SPPO1 molecules on graphene with the most stable adsorption configuration. The calculated $E_{ad.}$ values of different organic molecules on graphene surface are also shown. Red, yellow, cyan, and purple balls represent O, H, C and P atoms, respectively.

transfer of CVD-grown graphene films. Different from previously-used support materials, rosin has a super solubility in organic solvents, a weak interaction with graphene and sufficient strength, all of which enable clean and damage-free transfer. The transferred monolayer graphene films have a very low surface roughness with a maximum height of about 15 nm and a uniform sheet resistance of 560 Ω per square with about 1% deviation over a large area. Such clean and damage-free graphene greatly improves the current efficiency (CE) and power efficiency (PE) of OLEDs, with maxima as high as 89.7 cd A$^{-1}$ and 102.6 lm W$^{-1}$, respectively, even without doping. More importantly, we have fabricated the four-inch monolithic flexible graphene-based OLEDs with a uniform lighting area of 56 cm$^2$ and a high brightness of about 10,000 cd m$^{-2}$ that can already satisfy the requirements for lighting sources and displays.

## Results

**Basic principle of rosin as a support layer.** Based on the understanding of graphene transfer by using macromolecular

polymers and small organic molecules, an ideal polymeric support layer should satisfy all the following three requirements in order to obtain residue- and damage-free graphene films of a large area: (i) a good solubility in solvents, (ii) a low $E_{ad.}$ with the graphene surface, and (iii) sufficient support strength. Good solubility allows the support layer to be easily dissolved in the commonly used chemical solvents. A low $E_{ad.}$ is beneficial for the separation of the polymeric support layer from the graphene surface. Sufficient support strength can effectively prevent fragmentation or tearing of the graphene film during transfer.

Rosin is a small natural organic molecule polymer (molecular weight, ca. 302) mainly consisting of resin acids (primarily abietic acid)[44,45]. As discussed in Supplementary Note 2 and shown in Supplementary Table 3, it has a good solubility in commonly-used organic solvents such as alcohol, ether, acetone and chloroform. We performed density functional theory (DFT) calculations to evaluate the $E_{ad.}$ of rosin, pentacene, SPPO1 and HMMA ($C_{31}H_{52}O_{12}$, a very short chain of PMMA polymer for reducing computation complexity) molecules on a graphene

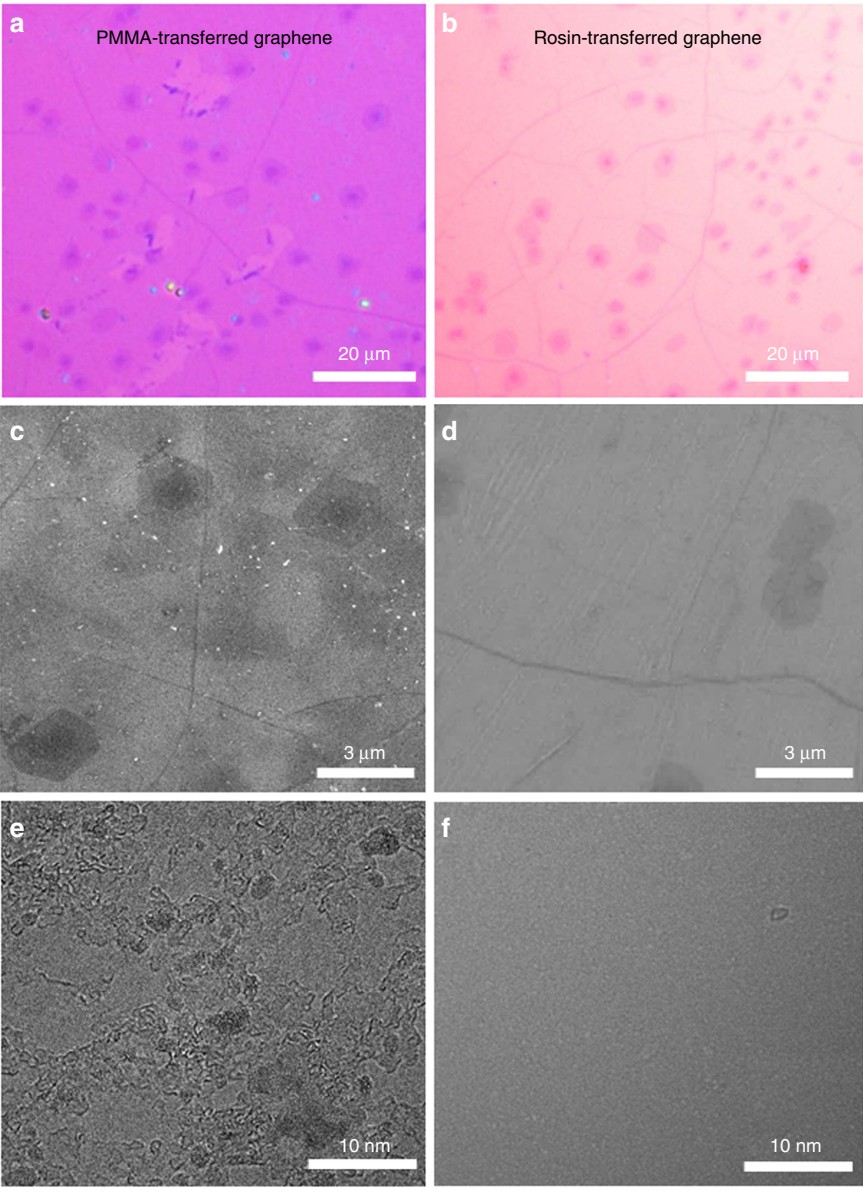

**Figure 2 | Surface structure characterization of graphene transferred using different support layers.** (**a,b**) OM, (**c,d**) SEM and (**e,f**) HRTEM images of (**a,c,e**) PMMA- and (**b,d,f**) rosin-transferred graphene films.

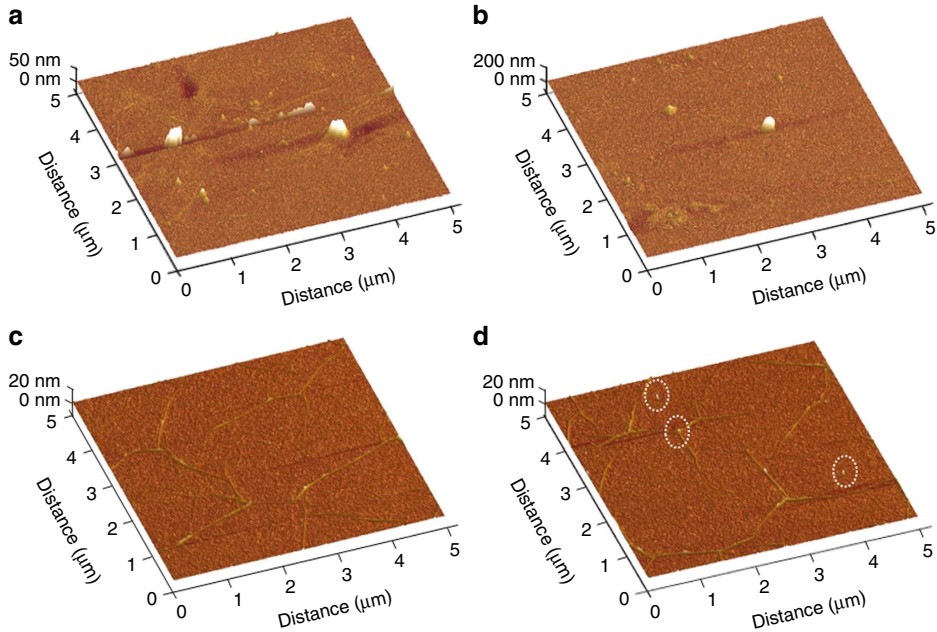

**Figure 3 | Surface roughness characterization of graphene transferred using different support layers.** Three-dimensional (3D) AFM images of (**a,b**) PMMA- and (**c,d**) rosin-transferred graphene films taken in areas (**a,c**) without large residue particles and in areas (**b,d**) with large residue particles. The white circles in **d** denote the rarely observed rosin residue particles.

surface (Supplementary Note 3). The DFT calculation results show that the $E_{ad.}$ (1.04 eV) of rosin on graphene with the most stable adsorption configuration (Fig. 1a) is about 1.4 times smaller than those of HMMA (1.45 eV, Fig. 1b) and pentacene (1.45 eV, Fig. 1c) molecules. It is reasonable to believe that the $E_{ad.}$ of PMMA on graphene is much larger than 1.45 eV because of its much longer chains. Although SPPO1 shows the smallest $E_{ad.}$ of the four polymers studied (Fig. 1d), it is too brittle to retain the integrity of graphene during transfer[39]. In contrast, a rosin layer is strong enough. As shown in Supplementary Figs 1 and 2b, neither macroscopic tears nor microscopic cracks can be found in the rosin layer after the sample was separated from a metal substrate and then collected on a target substrate. Therefore, rosin is expected to be an ideal support layer for the clean and damage-free transfer of CVD-grown graphene.

**Rosin-enabled ultraclean and damage-free transfer.** Supplementary Fig. 1 in Supplementary Note 4 shows the transfer process of a CVD-grown graphene film from a Cu foil by a substrate etching method with rosin as the support material. The graphene film is mostly monolayer with some bilayer or multi-layer islands on its surface (Supplementary Fig. 2a). The rosin support layer was prepared by spin coating a concentrated rosin solution (50 wt% rosin in ethyl lactate) with a high viscosity and good film-forming ability. We have also tried to use dilute rosin solution (20, 30 wt%) with low viscosity for transfer. However, the dilute rosin solution has poor film-forming ability and cannot form thick enough, smooth and uniform film to support the graphene (Supplementary Fig. 3). After the Cu foil was etched away, the graphene/rosin stack floating on the etchant solution was collected on a target substrate. Theoretically, rosin is freely soluble in acetone[44]. However, although the major component of rosin is resin acid, it also contains some other components such as dehydroabietic acid and its isomers. Therefore, we used acetone and banana oil solutions in sequence to remove the rosin layer.

We used optical microscopy (OM), scanning electron microscopy (SEM), high-resolution transmission electron microscopy (HRTEM), and atomic force microscopy (AFM) to characterize the surface of rosin-transferred graphene films. For comparison, the surface structure of PMMA-transferred graphene films was also studied. The individual dark dots shown in Fig. 2a–d are bilayer or multilayer graphene islands and the randomly distributed dark lines are wrinkles, which provide good indicators of the cleanness of the transferred graphene because of the strong absorptivity of wrinkles and graphene edges. Similar to results reported in the literature[34–37,46], a large number of PMMA particles are observed on the PMMA-transferred graphene even in low-magnification OM images (Fig. 2a,c,e). HRTEM images show that the graphene surface is covered by a nearly continuous thin layer of PMMA along with many particles (Fig. 2e), which is further confirmed by the decreased transparency (Supplementary Fig. 4). In sharp contrast, the rosin-transferred graphene films are ultraclean (Fig. 2b,d,f and Supplementary Fig. 5). No rosin residue is observed by SEM and AFM, even on the wrinkles and edges (Fig. 2d, Supplementary Fig. 5b,c), and only very few sparsely distributed tiny rosin residue particles are observed by HRTEM on the smooth surface of the graphene (Fig. 2f).

We then used AFM to characterize the surface roughness of the transferred graphene films (Fig. 3). 100 randomly selected areas ($5 \times 5 \, \mu m^2$) were measured for each sample. Consistent with OM observations, large residue particles are frequently observed on the PMMA-transferred sample (Fig. 3a,b), while only few small particles are occasionally observed on the rosin-transferred sample (Fig. 3c,d). The rosin-transferred graphene films show a small root mean square (RMS) roughness of 0.66 nm, which mainly originates from the wrinkles that are unavoidable for CVD-grown graphene on metals. This value is much lower than that of the PMMA-transferred graphene film (6.52 nm). Compared to RMS roughness, the maximum height ($R_{max}$) of the large residue particles is a more important parameter for large-area thin-film device applications, because this determines whether the devices are likely to be short-circuited. A typical statistical histogram of $R_{max}$ collected from 100 randomly selected areas ($5 \times 5 \, \mu m^2$) for a rosin-transferred graphene film on a $SiO_2/Si$ substrate was shown in Supplementary Fig. 6.

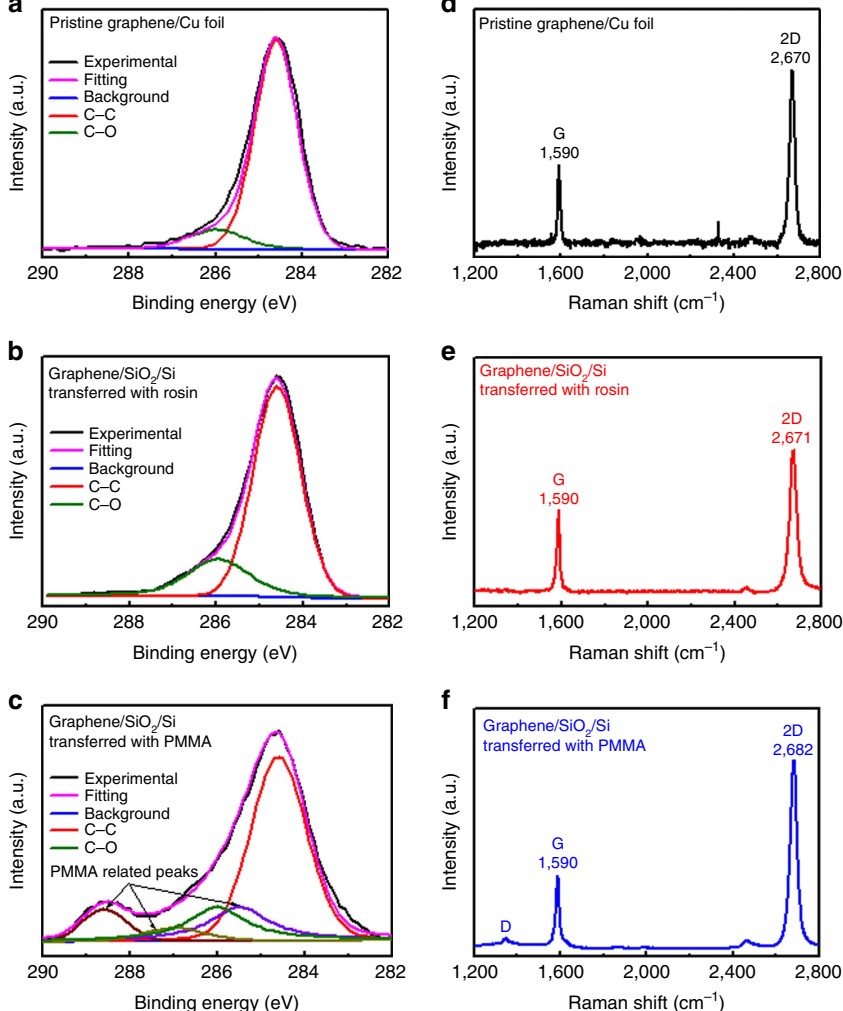

**Figure 4 | Chemical composition characterization of graphene transferred with different support layers.** (**a–c**) High-resolution C1s XPS spectra of (**a**) graphene/Cu foil, (**b**) graphene on SiO$_2$/Si transferred with rosin, (**c**) graphene on SiO$_2$/Si transferred with PMMA. The C-O peak (green) observed in the graphene on Cu foil is attributed to the adsorption of oxygen or water under ambient conditions. (**d–f**) Raman spectra of (**d**) graphene/Cu foil, (**e**) graphene on SiO$_2$/Si transferred with rosin and (**f**) graphene on SiO$_2$/Si transferred with PMMA.

Surprisingly, the rosin-transferred graphene film has a $R_{max}$ of about 15 nm (Fig. 3d and Supplementary Fig. 6), which is more than 10 times smaller than that of the PMMA-transferred graphene film (about 200 nm, Fig. 3b). Note that both the RMS roughness and $R_{max}$ of our PMMA-transferred monolayer graphene films is similar to those reported in the literature (Supplementary Table 4), which confirms the advantage of rosin as a support layer for the clean transfer of graphene.

X-ray photoelectron spectroscopy (XPS) is surface-sensitive quantitative spectroscopic technique that can measure the elemental composition of a surface in the parts-per-thousand range, and Raman spectroscopy provides a high-resolution characterization tool to give both the atomic structure and electronic properties of graphene such as the number and orientation of layers, doping, disorder and functional groups[47]. We have therefore used XPS and Raman spectroscopy to characterize the graphene transferred using rosin and PMMA support layers. As shown in Supplementary Figs 7 and 8, rosin and PMMA coatings on the as-grown graphene on Cu respectively lead to strong rosin- and PMMA-related XPS peaks and significant upshifts of the Raman 2D peak of about 15 and 30 cm$^{-1}$. However, the rosin-transferred graphene on a SiO$_2$/Si substrate shows almost the same XPS and Raman spectra as those

from the as-grown graphene on Cu (Fig. 4a,b,d,e). No rosin-related XPS and Raman peaks were detected as well as the defect-related D peak, confirming that the rosin has been effectively removed and no defects were generated during the transfer process. The small Raman 2D peak upshift of about 1 cm$^{-1}$ indicates that rosin-transferred graphene is slightly doped[48,49]. In contrast, as shown in Fig. 4c,f, the PMMA-transferred graphene shows similar XPS spectra to PMMA, a visible D peak and large 2D peak upshift of about 13 cm$^{-1}$ (a typical value for PMMA-transferred graphene reported in the literature[34,50,51]) although no PMMA-related Raman peaks are visible, suggesting that the graphene is strongly doped as well as having many PMMA residue particles on its surface.

We investigated the electrical and optical properties of graphene films transferred on PET substrates. Figure 5a shows a photograph of a rosin-transferred monolayer graphene film of size $10 \times 10$ cm$^2$, which was divided into 100 equal areas for electrical property measurements. All these areas show a very uniform sheet resistance of $560 \, \Omega$ per square with a standard deviation of about 1% (Fig. 5b) and a transmittance of about 97.4% at 550 nm wavelength (Supplementary Fig. 4). The small decrease in transmittance compared to ideal monolayer graphene (97.7%) is mainly attributed to the presence of a great number of

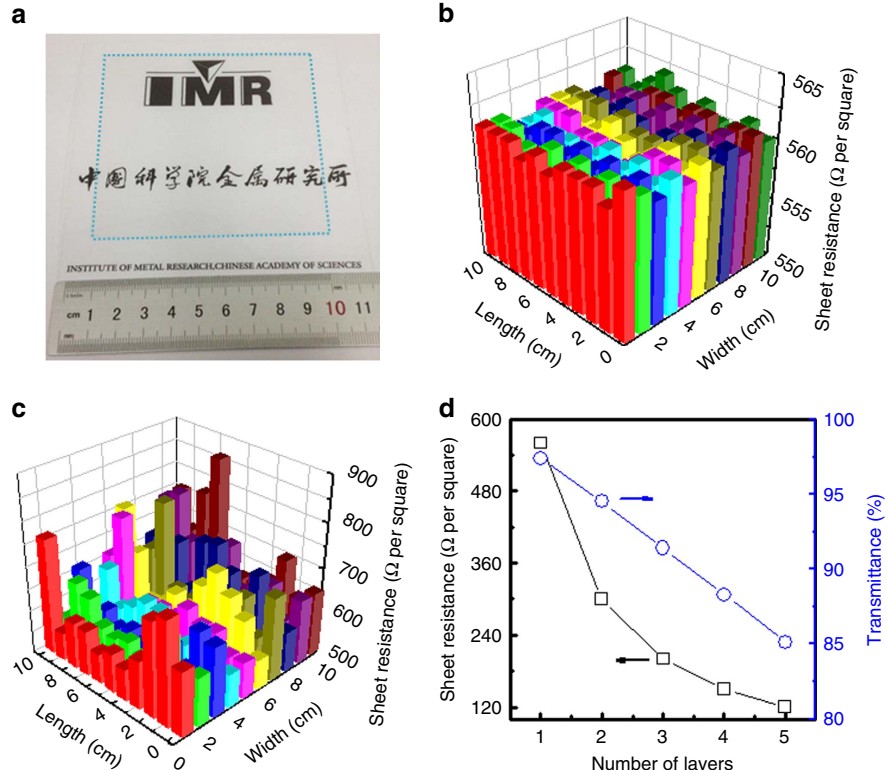

**Figure 5 | Electrical and optical properties of graphene transferred using different support layers.** (**a**) A $10 \times 10$ cm$^2$ monolayer graphene film (marked by blue dot square) transferred onto a PET substrate using rosin. (**b**) Sheet resistance map of the rosin-transferred monolayer graphene film in **a**. (**c**) Sheet resistance map of a $10 \times 10$ cm$^2$ PMMA-transferred monolayer graphene film. (**d**) Sheet resistance and transmittance at 550 nm wavelength of rosin-transferred graphene films with different number of layers.

small graphene islands on the monolayer surface (Fig. 2b,d)[52]. In contrast, the PMMA-transferred graphene film shows a higher sheet resistance of about $632\,\Omega$ per square with a large standard deviation of about 66% (Fig. 5c) and a lower transmittance of about 96.6% (Supplementary Fig. 4) although it is strongly p-doped according to Raman spectra analyses, indicating the presence of damage. The better electrical and optical properties give further evidence of the advantages of rosin over PMMA for the clean and damage-free transfer of large areas of graphene. In addition, the rosin-transferred graphene on PET has very good flexibility and little conductivity change on bending, with only a 10% increase in sheet resistance after bending 10,000 times to a radius of 2 cm (Supplementary Fig. 9).

In order to reduce the sheet resistance to meet the requirements of various electronic and optoelectronic applications, multilayer graphene films are usually fabricated by layer-by-layer transfer and stacking of monolayer graphene. As shown in Fig. 5d, when the rosin-transferred graphene film increases from monolayer to five layers, the transmittance decreases linearly from 97.4 to 85.1%, and the sheet resistance deceases from 560 to $120\,\Omega$ per square. Unfortunately, the roughness of graphene films is inevitably multiplied after stacking. As shown in Supplementary Fig. 10a, the RMS roughness and $R_{max}$ of the PMMA-transferred graphene films are greatly increased from about 6.52 and 200 nm for a monolayer to about 10.44 and 1,000 nm for five layers. Such a huge roughness far exceeds the typical thickness of the active layer of thin-film optoelectronic devices, and consequently causes a high leakage current and short circuiting. In contrast, the rosin-transferred five-layer graphene film is still very smooth, showing RMS roughness and $R_{max}$ values of about 3.51 and 35 nm (Supplementary Fig. 10b). The highly conductive smooth graphene films transferred with a rosin support layer open up the possibility for the fabrication of large-area flexible thin-film electronic and optoelectronic devices such as OLEDs and OPV cells.

**Large-area OLEDs with a rosin-transferred graphene anode.** We first used rosin-transferred three-layer graphene as an anode to fabricate phosphorescent green OLEDs with a lighting area of $0.4 \times 0.4$ cm$^2$, a typical device size reported in the literature. The structure and energy level diagram of the device are shown in Fig. 6a. As reported previously[53], in order to increase the work function of graphene and its compatibility with a hole-injection layer (HIL), we selectively oxidized the top layer by ozone treatment to form a graphene oxide (GO)/graphene (G) heterostructure anode. After that, we deposited a MoO$_3$ HIL, organic layers and cathode in sequence on the top of the GO/G anode to fabricate green OLED devices. For comparison, we also fabricated green OLEDs of the same size and device structure using a PMMA-transferred three-layer graphene film (the top layer was also selectively oxidized) and ITO as the anode.

As shown in Fig. 6b–d, both the CE and PE of the OLED with the rosin-transferred graphene anode are higher than those of devices using PMMA-transferred graphene and ITO as anodes at the same operating voltage. The maximum CE and PE of OLEDs with the rosin-transferred graphene anode can reach 89.7 cd A$^{-1}$ and 102.6 lm W$^{-1}$, respectively, which are comparable to the best values of graphene-based OLEDs reported in the literature without any outcoupling structure and cavity resonance enhancement design (Supplementary Table 1). Moreover, it is necessary to point out that our graphene anode is very stable. In contrast, the reported OLEDs with comparable performance usually use graphene films doped by HNO$_3$ or AuCl$_3$ as the anode[12,13,54,55], and these are very unstable and can greatly degrade

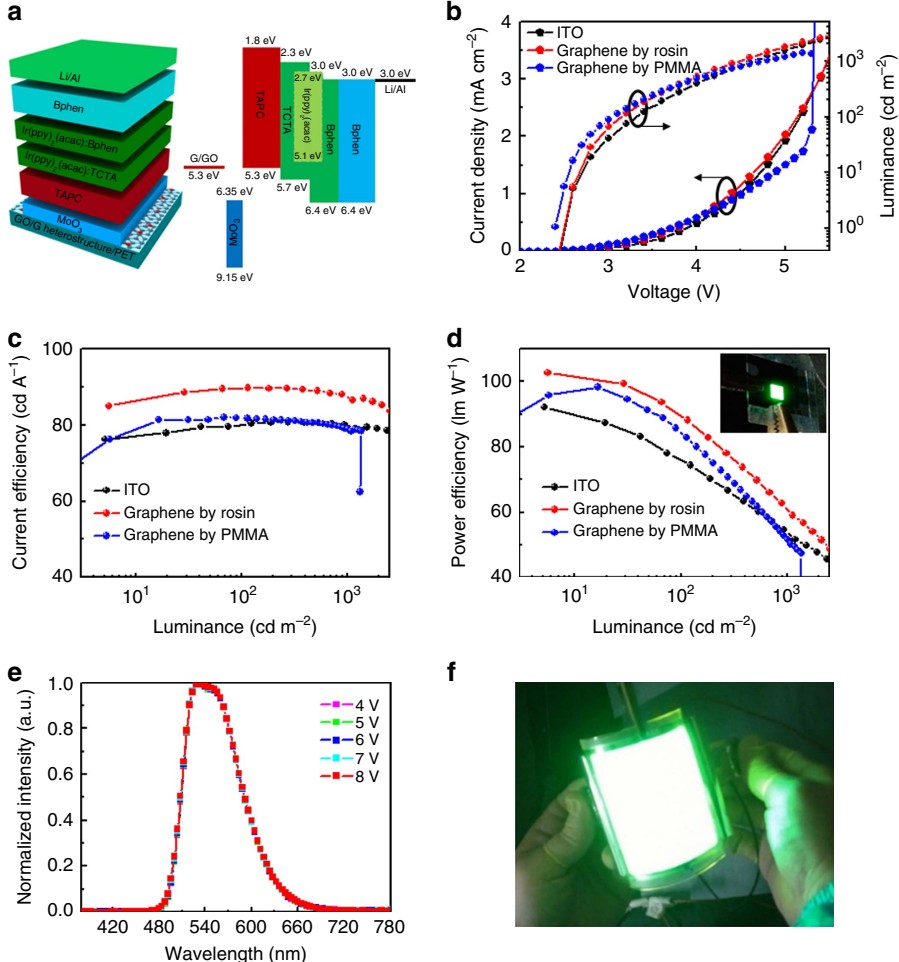

**Figure 6 | Device structure and performance of green OLEDs with different anodes.** (**a**) Device structure (left) and energy level diagram (right). (**b**) Current-voltage characteristics, (**c**) CE and (**d**) PE versus luminance characteristics of OLEDs with rosin-transferred 3-layer graphene, PMMA-transferred 3-layer graphene and ITO films as anodes. (**e**) Normalized electroluminescence spectra obtained at different voltages. (**f**) A four-inch monolithic flexible green OLED with a rosin-transferred five-layer graphene anode, showing uniform luminance and excellent flexibility.

device efficiency and lifetime. The identical electroluminescence spectra obtained at different voltages indicate the good stability of our devices (Fig. 6e). Although the OLED with the PMMA-transferred graphene anode shows a higher current density than the device with the ITO anode at low voltage, severe current leakage occurs and both CE and PE decrease quickly along with the burn-out of the device at high driving voltages. This is attributed to the huge surface roughness of the PMMA-transferred graphene anode as shown above. Moreover, it is necessary to point out that the yield of OLEDs with rosin-transferred graphene TCEs is about 100%, while the yield of OLEDs with PMMA-transferred graphene TCEs is lower than 50%, although the device area is only $0.4 \times 0.4\,cm^2$.

The OLED device failure induced by large surface roughness becomes more serious as the device size is increased. Therefore, the lighting area of the OLEDs with graphene TCEs reported in the literature is mostly limited to less than $1\,cm^2$ (Supplementary Table 1). To further show the advantages of our rosin transfer method, we tried to fabricate large-area OLEDs with rosin- and PMMA-transferred graphene as anodes. Here five-layer graphene films were used as TCEs to further reduce the sheet resistance. Figure 6f and Supplementary Video 1 show a four-inch monolithic flexible green OLED with a lighting area of $8 \times 7\,cm^2$ fabricated with a rosin-transferred graphene anode. The whole device can be lighted at about 5 V, and the brightness

increases with increasing applied voltage. It is worth noting that the luminescence is very uniform over the whole four-inch lighting area at a fixed voltage. At an applied voltage of 16 V, the OLED device shows a brightness in excess of $10,000\,cd\,m^{-2}$, which already satisfies the requirements of lighting sources and displays and is even better than some commercial-off-the-shelf OLED panels. In addition, the devices have very good flexibility because of the excellent electromechanical stability of the rosin-transferred graphene films (Supplementary Video 1). No luminous intensity change was observed after repeatedly bending tens of times even at a high voltage of 16 V.

In contrast, for the PMMA-transferred graphene TCEs, no four inch OLED devices can be lighted. As shown in Supplementary Video 2, although some three inch OLEDs can be lighted gradually from the edge to the center, the center area is hardly lighted and no uniform luminescence is observed. In addition, all the devices break down quickly. One possible reason is that the sheet resistance of a PMMA-transferred 5-layer graphene film is relatively high (about $200\,\Omega$ per square), thus it is difficult for current to flow from the edge contacting the electrode to the center. In addition, the insulating PMMA residue is an inhibitor for charge extraction, resulting in non-uniform lighting. More seriously, the $R_{max}$ (about 1,000 nm) of the large PMMA residue particles far exceeds the thickness of the active layer (about 140 nm), resulting in electrical micro-short circuits. Therefore,

many dark spots are observed and these grow quickly with increasing operating time and applied voltage, and spark discharge can be found at some spots of the lighted area, which lead to quick burn-out of the device.

We also compared the stability of large-area and small-area OLED devices. The unencapsulated $0.4 \times 0.4 \, cm^2$ OLEDs can light more than two times longer than four-inch ones ($8 \times 7 \, cm^2$) in air at same luminance. The main reason is that the large-area devices easily exist further hole defects that cause short circuit, thus reducing stability. Another important problem is that the anode resistance has much larger influence on the large-area devices than the small-area ones. It can cause a big voltage drop and current distribution non-uniformity in large-area devices, which will greatly reduce device stability. Therefore, for our large-area OLED devices using rosin-transferred graphene as anode, much effort should be made to further reduce its resistance to enhance device lifetime in the future.

## Discussion

Ultraclean and damage-free transfer of large-area CVD-grown graphene films has been achieved using the small organic molecule rosin as a support layer based on its good solubility, weak interaction with graphene and adequate support strength. The transferred graphene films have a very low surface roughness with a maximum height of residue particles up to 15 nm and an extremely uniform sheet resistance of $560 \, \Omega$ per square with about 1% deviation over a large area. Such clean and damage-free graphene greatly improves the CE and PE of OLEDs, and more importantly, it has enabled to production of the 4-inch monolithic flexible graphene-based OLEDs exhibiting uniform light emission and a high brightness of about $10,000 \, cd \, m^{-2}$. This rosin-based transfer method provides a universal approach for the ultraclean and damage-free transfer of graphene and other 2D materials grown by CVD on metals, which paves the way for electronic and optoelectronic applications, in particular, large-area thin-film devices of 2D materials.

## Methods

**Theoretical calculations of adsorption energy.** DFT calculations were performed using the projector augmented wave method[56,57] and a plane-wave (PW) basis set as implemented in the Vienna *ab-initio* simulation package[58]. The Perdew-Burke-Ernzerhof functional[59] for the exchange-correlation term was used for all calculations. The energy cutoff for the PW basis set was set to be 400 eV. A large periodic and orthorhombic graphene supercell ($21.30 \times 19.68 \times 30 \, Å^3$) was used to calculate the adsorption energies of different organic molecules on the graphene surface. Only the $\Gamma$ point was used to sample the first Brillouin zone for all calculations due to the large size of the graphene supercell. For the geometry relaxations and energy calculations, van der Waals interactions were incorporated by the optB88 exchange functional[60,61], which has been proved to be very important to accurately evaluate the interactions between molecules and/or clusters on a graphene surface[62]. All atoms are allowed to be fully relaxed in the fixed $21.30 \times 19.68 \times 30 \, Å^3$ supercell until the residual force per atom decreases to $<0.01 \, eV \, Å^{-1}$.

**Fabrication and transfer of graphene.** Monolayer graphene films were grown by CVD on copper foils as reported previously[23,24,63]. Typically, a roll of 25 μm-thick copper foil (99.9%, Shanghai, China) was first annealed at 1,000 °C under a 5 sccm hydrogen flow in a 3-inch-wide tubular quartz reactor, and then exposed to the mixture of hydrogen (5 sccm) and methane (35 sccm) at a total pressure of 100 Pa for 30 min to grow graphene, which is followed by a slow cooling process to room temperature. After growth, the graphene films were transferred to the target substrate following the scheme shown in Supplementary Fig. 1. Typically, a thin layer of rosin (average $M_w$ *ca.* 302 by gel permeation chromatogram, Alfa-Aesar CAS no. 8050-09-7, dissolved in ethyl lactate with a concentration of 50 wt.%) was first spin-coated on the CVD-grown graphene film at 500 r.p.m. for 10 s and then at 1,200 r.p.m. for 60 s. The rosin layer was then cured at room temperature, followed by etching the Cu foil in an aqueous solution of $FeCl_3$ ($0.03 \, g \, ml^{-1}$) to obtain a rosin/graphene stack floating on the solution. After washing with deionized water to remove residual etchant, the rosin/graphene stack was collected on the target substrate, and then taken out from the solution. To ensure the rosin/graphene film stack remained intact and was fully in contact with the target substrate,

the rosin/graphene/target substrate was first treated at 40 °C for 1 h, and the temperature then was slowly increased to 120 °C for 20 min to evaporate the residual water. Subsequently, the rosin layer was dissolved by acetone (Analytical reagent, 99%) and banana oil solutions (Analytical reagent, 99%) in sequence. Finally, the graphene film was blow dried using high-purity nitrogen.

**Fabrication of flexible OLED devices.** First, three-layer graphene films were transferred layer-by-layer onto a PET substrate using rosin as the support layer. Ozone treatment was carried out at 120 °C for 5 min to obtain a GO/G/G heterostructure to simultaneously increase the work function and compatibility with HIL as described in our earlier work[53]. GO/G/G heterostructure electrodes were then patterned by covering with a shadow mask and subsequently rubbing away the uncovered area. ITO anodes with the same patterns were cleaned by acetone, alcohol, and deionized water, followed by UV/ozone treatment. The graphene-based anodes and ITO anodes were then loaded into a high vacuum chamber for the deposition of a 5 nm $MoO_3$ HIL layer. After that, phosphorescent green OLEDs were fabricated by subsequently depositing a 60 nm di-[4-(N,N-ditolyl-amino)-phenyl] cyclohexane (TAPC) hole transportation layer (HTL), two 8 nm layers of bis(2-phenylpyridine) (acetylacetonate)iridium(III) [Ir(ppy)$_2$(acac)] doped with 1,1-bis[4-[N,N-di(p-tolyl)amino]phenyl] cyclohexane (TCTA) and a bathophenanthroline (Bphen) light emission layer, a 60 nm Bphen electron transportation layer (ETL) and a 0.5 nm Li/130 nm Al cathode. The active area defined by the cathode is $0.4 \times 0.4 \, cm^2$. For comparison, PMMA-transferred 3-layer graphene films were also prepared, selectively oxidized and patterned by the same methods, and fabricated into OLEDs with the same device structure. For the fabrication of large-area devices, rosin- and PMMA-transferred 5-layer graphene films were used to further reduce the sheet resistance, while the other fabrication procedures and device structures remained the same.

**Characterization.** OM (Nikon Eclipse LV100), SEM (Nova NanoSEM 430, acceleration voltage of 15 kV) and HRTEM (FEI TECNAI G2 F20, acceleration voltage of 120 kV) were used to characterize the morphology and structure of the graphene films transferred onto a $SiO_2$ (300 nm thick)/Si substrate and a TEM grid. AFM (Dimension Icon, Bruker, Inc.) was used to characterize the surface roughness of the graphene transferred onto the $SiO_2$/Si substrate with a tapping mode. Particle analysis function in NanoScope Analysis 1.40, a software package for analysing scanning probe microscopy data, was used to analyse the height of the residue particles to obtain the $R_{max}$ of each measured area. $R_{max}$ of a transferred graphene sample is the maximum of the $R_{max}$ values of all measured areas. XPS was used to characterize the surface chemical composition on an ESCALAB 250 instrument with Al $K_\alpha$ and He I radiation sources. The XPS spectra were fitted using the XPS peak 4.1 software in which a Shirley background was assumed. Raman spectra were measured using a Jobin Yvon LabRam HR800, excited by a 532 nm laser. The laser spot size was about 1 μm with the laser power below 2 mW to avoid laser-heating-induced sample damage.

The sheet resistance and transmittance of graphene films with different numbers of layers transferred onto PET were measured by a 4-probe resistivity measurement system (RTS-9, Guangzhou, China) and UV–vis-NIR spectrometer (Agilent Model Cary 5E), respectively. Current-brightness-voltage characteristics of the unencapsulated OLEDs were characterized by Keithley source measurement units (Keithley 2400 and Keithley 2000) with a calibrated silicon photodiode in air. Note that our OLED devices were stable enough when measuring their basic optical and electric properties although they were not encapsulated.

**Data availability.** The data that support the findings of this study are available from the corresponding author upon request.

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

## Acknowledgements

This work is supported by the Ministry of Science and Technology of China (No. 2016YFA0200101), National Science Foundation of China (Nos. 51325205, 51290273, 51521091, 11661131001 and 51572265), Chinese Academy of Sciences (KGZD-EW-303-1, KGZD-EW-303-3, and KGZD-EW-T06), and Postdoctoral Science Foundation of China (No. 2015M580237). We thank M. Li, Y. Sun, Z.B. Liu, T. Ma, D.M. Sun and L.B. Gao for their kind help for structure characterization and device performance measurements.

## Author contributions

W.R. proposed and supervised the project; Z.Z., J.D. and W.R. designed the experiments; Z.Z. performed the experiments with the help of D.Z. and L.M.; H.S. and J.C. performed OLED measurements under the supervision of D.M.; L.Y. performed theoretical calculations; W.R., J.D. and Z.Z. analysed the data; W.R., J.D., Z.Z. and H.-M.C. wrote the manuscript. All the authors discussed the results and commented on the manuscript.

## Additional information

**Competing financial interests:** The authors declare no competing financial interests.

