## [Peer Review File · Nature Communications]

Reviewers' comments:

Reviewer #1 (Remarks to the Author):

Large-area, clean surface transfer of graphene is essential to its applications in electronics and optoelectronics. The most widely used method at present, i.e. PMMA-assisted transfer method, suffers from the large amount of surface PMMA residue. In this work, Zhang et al. developed a rosin-assisted method for large-area, ultraclean, and damage-free transfer of graphene. Comprehensive characterizations on the rosin-transferred graphene were performed to demonstrate the high-quality of the graphene layer. The ultraclean surface, high conductivity, and high transmittance of the graphene layers enabled the fabrication of 4-inch OLEDs with remarkably improved efficiency and uniformity. This work should be of interest to the graphene research community. I recommend the publication of this manuscript after addressing the following questions:

- (1) Have the authors assessed the lifetime of the OLEDs fabricated from the rosin-transferred graphene? Please compare the large-area devices with small-area devices.
- (2) 50 wt% rosin in ethyl lactate was used in this study. What's the influence of rosin concentration?

Reviewer #2 (Remarks to the Author):

"Rosin-enabled ultraclean and damage-free transfer of graphene for large-area flexible organic light-emitting diodes"
Ren et al.

NCOMMS: What are the major claims of the paper?

The authors claim a method to transfer large-area, high-quality graphene sheets for organic semiconductor devices, such as OLEDs. The central claims to the large-area, high-quality process are:

- Small-molecule max residue height 15 nm
- High quality transfer, as evidenced by OM, SEM, AFM and XPS analysis
- Single sheet resistance 560 ohm/square
- 4-inch monolithic OLED with improved current and power efficiencies

NCOMMS: Are they novel and will they be of interest to others in the community and the wider field? If the conclusions are not original, it would be helpful if you could provide relevant references.

The author's method and claims are relevant to scientists and engineers interested in graphene, organic electronics, and OLEDs. The experimental results, analysis, and OLED devices produced are complete and excellent.

The authors should cite the previous work by Bae et al. on large-area graphene films, and demonstrate why their method is novel compared to the prior work:

Bae, Sukang, et al. "Roll-to-roll production of 30-inch graphene films for transparent electrodes." Nature nanotechnology 5.8 (2010): 574-578.

During a search for prior work, I found a related patent 'Transferring method of graphene using self-adhesive film', published Nov. 5, 2015 (US 20150314579 A1 / WO 2014109619 A1). The prior publication of a similar rosin process may reduce the novelty of this work.

NCOMMS: Is the work convincing, and if not, what further evidence would be required to strengthen the conclusions?

The authors analysis of graphene-transfer support layers based on adsorption energy is a valuable tool (sp², sp³, pi-pi interactions). The authors present a well-thought-out and reasoned motivation for the investigation. The demonstration of stacking multiple layers of graphene to reduce sheet resistance is vital for use in organic optoelectronic devices; the surface roughness remains low despite the addition of multiple layers.

NCOMMS: On a more subjective note, do you feel that the paper will influence thinking in the field? Please feel free to raise any further questions and concerns about the paper.

Lines 178-179: Surprisingly, the rosin-transferred graphene film has a R_{max} of ~15 nm (Fig. 3d), which is more than 10 times smaller than that of the PMMA-transferred graphene film (~200 nm, Fig. 3b).

I would like the authors to propose a mechanism for the low R_{max} found in the rosin transfer process. Is the low R_{max} value repeatable? What are the statistics of the process? The authors report that 100 5x5 um areas were used for this experiment; I suggest a histogram or similar plot to show the R_{max} statistics.

NCOMMS: We would also be grateful if you could comment on the appropriateness and validity of any statistical analysis, as well the ability of a researcher to reproduce the work, given the level of detail provided.

The authors demonstrate excellent large-area (4") OLED devices. I request that the authors detail any encapsulation used in their OLED device stack; it is my understanding that the OLED materials used are not stable in air, and therefore will not survive operation in an O₂/H₂O atmosphere.

I request that the authors specify the process and Cu foil material used for graphene growth. The citation [53] does not adequately describe the process to create large area graphene on Cu foil.

Best wishes,

Jonathan Beck, PhD

Response to the reviewers' comments

Reviewer #1

Large-area, clean surface transfer of graphene is essential to its applications in electronics and optoelectronics. The most widely used method at present, i.e. PMMA-assisted transfer method, suffers from the large amount of surface PMMA residue. In this work, Zhang et al. developed a rosin-assisted method for large-area, ultraclean, and damage-free transfer of graphene. Comprehensive characterizations on the rosin-transferred graphene were performed to demonstrate the high-quality of the graphene layer. The ultraclean surface, high conductivity, and high transmittance of the graphene layers enabled the fabrication of 4-inch OLEDs with remarkably improved efficiency and uniformity. This work should be of interest to the graphene research community. I recommend the publication of this manuscript after addressing the following questions:

Response: We thank the reviewer very much for positive comments.

(1) Have the authors assessed the lifetime of the OLEDs fabricated from the rosin-transferred graphene? Please compare the large-area devices with small-area devices.

Response: We thank the reviewer very much for kind suggestion.

As we know, the stability of OLEDs is affected by many factors, including material stability, device structure, transport and injection balance of electrons and holes, and interface stability between organic/organic, organic/inorganic and inorganic/electrode, and the OLEDs have to be encapsulated during lifetime testing in order to keep it from O₂/H₂O in air. Moreover, the main point of this paper is to show that rosin allows ultraclean and damage-free transfer of graphene films, which consequently enables the fabrication of large-area graphene-based OLEDs. Therefore, we did not assess the lifetime of the OLEDs, which is beyond the consideration in this paper.

Generally, the small devices have longer lifetime than the large ones. According to the reviewer's suggestion, we have compared the stability of the large-area and small-area unencapsulated OLED devices. The small-area OLEDs ($0.4 \times 0.4 \text{ cm}^2$) can light more than two times longer than the large-area ones ($8 \times 7 \text{ cm}^2$) in air at the same luminance. The main reason is that the large-area devices easily exist further hole defects that cause short circuit, thus reducing stability. Another important problem is that the anode resistance has much larger influence on the large-area devices than the small-area ones. It can cause a big voltage drop and current distribution non-uniformity in large-area devices, which will greatly reduce device stability. For our large-area devices with rosin-transferred graphene as anode, we have to further reduce its resistance to enhance device lifetime, which will be done in the future.

We have added the above discussions in the revised manuscript.

(2) 50 wt% rosin in ethyl lactate was used in this study. What's the influence of rosin concentration?

Response: The rosin concentration plays an important role in the integrity of the transferred graphene film. We have tried to use different concentrations of rosin solutions (20, 30, 50 wt% rosin in ethyl lactate) for transfer. It was found that the dilute rosin solution with low viscosity has poor film-forming ability and cannot form thick enough, smooth and uniform film to support the graphene. Thus the graphene film is easily destroyed during transfer process as shown in Figure R1. Therefore, saturated rosin solution (50 wt% rosin in ethyl lactate) was used in our experiments to keep intact transfer of graphene films.

We have added the above discussions in the revised manuscript.

Figure R1. Transfer of a CVD-grown graphene film (1 cm^2) from a Cu foil to a SiO_2/Si substrate using a spin-coated rosin film (20 wt% rosin in ethyl lactate). (a) A floating rosin/graphene stack in DI water after removing the Cu foil by FeCl_3 etching, showing obvious tearing. (b) A rosin/graphene stack collected on a SiO_2/Si substrate, showing obvious damages of the transferred graphene film.

Reviewer #2

NCOMMS: What are the major claims of the paper?

Reviewer: The authors claim a method to transfer large-area, high-quality graphene sheets for organic semiconductor devices, such as OLEDs. The central claims to the large-area, high-quality process are:

- Small-molecule max residue height 15 nm
- High quality transfer, as evidenced by OM, SEM, AFM and XPS analysis
- Single sheet resistance 560 ohm/square
- 4-inch monolithic OLED with improved current and power efficiencies

Response: We thank the reviewer very much for the accurate summary of the major claims of our manuscript.

NCOMMS: Are they novel and will they be of interest to others in the community and the wider field? If the conclusions are not original, it would be helpful if you could provide relevant references.

Reviewer: The author's method and claims are relevant to scientists and engineers interested in graphene, organic electronics, and OLEDs. The experimental results, analysis, and OLED devices produced are complete and excellent.

Response: We thank the reviewer very much for positive comments.

Reviewer: The authors should cite the previous work by Bae et al. on large-area graphene films, and demonstrate why their method is novel compared to the prior work: Bae, Sukang, et al. "Roll-to-roll production of 30-inch graphene films for transparent electrodes." *Nature nanotechnology* 5.8 (2010): 574-578.

Response: We thank the reviewer very much for kind suggestion.

Bae et al. developed a very promising roll-to-roll transfer method with thermal release tape as support layer, which enables the production of 30 inch graphene transparent conductive films for the first time. However, a large number of particle-like adhesive residues and cracks are usually observed especially for the first transferred graphene layer, as shown in the SEM and AFM images in Fig. S4 and S5 (*Nature Nanotechnology* 5 (2010) 574-578). As the authors proposed, this is because the adhesion force between graphene and PET competes with the force between graphene thermal release tapes. Although the additional transfers look cleaner because the adhesion forces of thermal release tapes are smaller than graphene-graphene adhesion, large particle-like residues still can be observed obviously. In sharp contrast, our rosin-transferred graphene films are ultraclean and damage free, no big rosin residues or cracks are observed by SEM and AFM (Figs. 2b, d and S5 in our manuscript).

According to the reviewer's suggestion, we have cited this work and summarized the above differences in the revised manuscript.

Reviewer: During a search for prior work, I found a related patent ‘Transferring method of graphene using self-adhesive film’, published Nov. 5, 2015 (US 20150314579 A1 / WO 2014109619 A1). The prior publication of a similar rosin process may reduce the novelty of this work.

Response: We thank the reviewer very much to point out this patent. The patent discloses a transferring method of graphene using a self-adhesive film, which can be reused after graphene is transferred. In the patent, the inventors point out that by way of example, the self-adhesive film may include rosin as a tackifier resin. In contrast, in our method, the rosin is used independently as a support layer to prevent the graphene film from cracking during the transfer process. After the graphene is transferred onto target substrate, the rosin film needs to be removed by washing. Therefore, the roles of rosin are totally different for the present method and that disclosed in the patent. As a result, the transferred graphene films show dramatically different surface roughness and electrical conductivity. As shown in Figure 2-5 in the patent, there are still many particle-like polymer materials (up to hundred nanometers in height) derived from the self-adhesive film left on the surface of the transferred graphene films. The transferred film shows sheet resistance of about 975 ohm sq^{-1} . In our method, however, the weak interaction with graphene, good solubility and sufficient strength of rosin enable ultraclean and damage-free transfer of graphene. The transferred graphene has a very low surface roughness with an occasional maximum residue height of $\sim 15 \text{ nm}$ and an extremely uniform sheet resistance of 560 ohm sq^{-1} . In addition, for the patented transfer method, any one or both of the first and the second substrate have at least one characteristic of transparency, flexibility, and stretchability. However, our rosin-based transfer method has no requirement in transparency, flexibility, and stretchability for the first and the second substrates, unless it is used for the fabrication of flexible transparent conductive films.

We have cited this patent and summarized the above differences in the revised manuscript.

NCOMMS: Is the work convincing, and if not, what further evidence would be required to strengthen the conclusions?

Reviewer: The authors analysis of graphene-transfer support layers based on adsorption energy is a valuable tool (sp², sp³, pi-pi interactions). The authors present a well-thought-out and reasoned motivation for the investigation. The demonstration of stacking multiple layers of graphene to reduce sheet resistance is vital for use in organic optoelectronic devices; the surface roughness remains low despite the addition of multiple layers.

Response: We thank the reviewer very much for kind comment.

NCOMMS: On a more subjective note, do you feel that the paper will influence thinking in the field? Please feel free to raise any further questions and concerns about the paper.

Reviewer: Lines 178-179: Surprisingly, the rosin-transferred graphene film has a R_{max} of ~15 nm (Fig. 3d), which is more than 10 times smaller than that of the PMMA-transferred graphene film (~200 nm, Fig. 3b). I would like the authors to propose a mechanism for the low R_{max} found in the rosin transfer process. Is the low R_{max} value repeatable? What are the statistics of the process? The authors report that 100 5x5 um areas were used for this experiment; I suggest a histogram or similar plot to show the R_{max} statistics.

Response: Thank you very much for good suggestions.

R_{max} is the maximum height of the polymer residue particles generated during transfer. As we mentioned in the manuscript, rosin is a small natural organic molecule and it has good solubility in solvents and low adsorption energy ($E_{ad.}$) with the graphene surface. In contrast, PMMA has a large $E_{ad.}$ with graphene films and low solubility in any known organic solvent. The good solubility allows the rosin layer to be easily dissolved in the commonly used chemical solvents. The low $E_{ad.}$ is beneficial for the separation of the rosin layer from the graphene surface. As a result, the rosin-transferred graphene is ultraclean. Nearly no rosin residues are observed by

SEM and AFM, even on the wrinkles and edges (Figs. 2b, d, and S5), and only very few sparsely distributed tiny rosin residue particles are observed by HRTEM (Fig. 2f). This is the reason why the rosin-transferred graphene has a very low R_{max} .

Figure R2. Typical 3D AFM images of another rosin-transferred graphene sample.

The low R_{max} value is well repeatable. Figure R2 shows 3D AFM images taken from another rosin-transferred graphene sample, showing similar surface roughness with those shown in Fig. 3c, d in the main text. For the statistics of R_{max} , AFM (Dimension Icon, Bruker Inc) was first used to characterize the surface of the graphene films transferred on a SiO_2/Si substrate with a tapping mode. 100 randomly selected areas ($5 \times 5 \mu\text{m}^2$) were measured for each sample. Then, we used particle analysis function in NanoScope Analysis 1.40, a software package for analyzing scanning probe microscopy data, to analyze the height of the residue particles to obtain the R_{max} of each area. R_{max} of a transferred graphene sample is the maximum of the R_{max} values of all measured areas. Figure R3 shows a typical statistical histogram of R_{max} collected from 100 randomly selected areas ($5 \times 5 \mu\text{m}^2$) for a rosin-transferred graphene film on a SiO_2/Si substrate, indicating that the rosin-transferred graphene film has a low R_{max} of ~ 15 nm.

We have discussed the mechanism for the low R_{max} of the rosin-transfer samples more clearly and given the statistics process of the R_{max} in detail in the revised manuscript. The statistical histogram of R_{max} has also been added in the revised Supplementary Information.

Figure R3. Histogram of R_{max} collected from 100 randomly selected areas ($5 \times 5 \mu\text{m}^2$) of a rosin-transferred graphene film on a SiO_2/Si substrate.

NCOMMS: We would also be grateful if you could comment on the appropriateness and validity of any statistical analysis, as well the ability of a researcher to reproduce the work, given the level of detail provided.

Reviewer: The authors demonstrate excellent large-area (4") OLED devices. I request that the authors detail any encapsulation used in their OLED device stack; it is my understanding that the OLED materials used are not stable in air, and therefore will not survive operation in an $\text{O}_2/\text{H}_2\text{O}$ atmosphere.

Response: Thank you very much for your kind suggestion. We did not encapsulate our OLEDs. They are stable enough when measuring their basic optical and electric properties in air. We can also measure them in N_2 glovebox, and we did not find large difference between two cases. As the reviewer mentioned, OLEDs are unstable in air due to $\text{O}_2/\text{H}_2\text{O}$. They have to be encapsulated when studying their lifetime, which is beyond the scope of this paper.

We have described the characterization of OLEDs more clearly in the revised manuscript.

Reviewer: I request that the authors specify the process and Cu foil material used for graphene growth. The citation [53] does not adequately describe the process to create large area graphene on Cu foil.

Response: We thank the reviewer very much for kind suggestions. Large-area graphene films were grown by CVD on copper foils in our group. Typically, a roll of 25 μm -thick copper foil (99.9%, Shanghai, China) was first annealed at 1000 $^{\circ}\text{C}$ under a 5 sccm hydrogen flow in a 3-inch-wide tubular quartz reactor, and then exposed to the mixture of hydrogen (5 sccm) and methane (35 sccm) at a total pressure of 100 Pa for 30 min to grow graphene, which is followed by a slow cooling process to room temperature.

We have added the above information and cited the following 3 papers related to large area graphene growth on Cu foils in the revised manuscript.

[1] Bae, S. *et al.* Roll-to-roll production of 30-inch graphene films for transparent electrodes. *Nat. Nanotechnol.* **5**, 574-578 (2010).

[2] Li, X. S. *et al.* Large-area synthesis of high-quality and uniform graphene films on copper foils. *Science* **324**, 1312-1314 (2009).

[3] Gao, L. B. *et al.* Efficient growth of high-quality graphene films on Cu foils by ambient pressure chemical vapor deposition. *Appl. Phys. Lett.*, **97**, 183109(1-3) (2010).

REVIEWERS' COMMENTS:

Reviewer #1 (Remarks to the Author):

The authors have satisfactorily addressed all the questions arising by the reviewer. I recommend the acceptance of this manuscript as it is.

Reviewer #2 (Remarks to the Author):

I appreciate the thoughtful discussions and citations that the authors have added to their manuscript. I recommend the manuscript for publication.

Best wishes.

Response to the reviewers' comments

Reviewer #1

The authors have satisfactorily addressed all the questions arising by the reviewer. I recommend the acceptance of this manuscript as it is.

Response: We thank the reviewer very much for positive comments.

Reviewer #2

I appreciate the thoughtful discussions and citations that the authors have added to their manuscript. I recommend the manuscript for publication.

Response: We thank the reviewer very much for positive comment.